# Respiration rates of marine prokaryotes and implications for the in vivo INT method

Isabel Seguro<sup>1</sup>, Kevin Vikström<sup>1,2</sup>, Jonathan D. Todd<sup>3,4,5</sup>, Stephen J. Giovannoni<sup>6</sup>, E. Elena García-Martín<sup>1,7</sup>, Robert Utting<sup>1</sup>, Carol Robinson<sup>1</sup>

- Ocean and Atmospheric Sciences, School of Environmental Sciences, University of East Anglia, Norwich, UK Department of Ecology and Genetics/Limnology, Uppsala University, Uppsala, Sweden
  - <sup>3</sup>School of Biological Sciences, University of East Anglia, Norwich, UK
  - <sup>4</sup>MOE Key Laboratory of Evolution and Marine Biodiversity, Frontiers Science Center for Deep Ocean Multispheres and Earth System & College of Marine Life Sciences, Ocean University of China, Qingdao, China
- 10 <sup>5</sup>Centre for Microbial Interactions, Norwich Research Park, Norwich, UK
  - <sup>6</sup>Department of Microbiology, Oregon State University, Corvallis, USA
  - <sup>7</sup>Ocean BioGeosciences, National Oceanography Centre, Southampton, UK

Correspondence to: Isabel Seguro (i.seguro-requejo@uea.ac.uk)

Abstract. The balance between the uptake of CO<sub>2</sub> by phytoplankton photosynthesis and the production of CO<sub>2</sub> from prokaryoplankton, zooplankton and phytoplankton respiration controls how much carbon can be stored in the ocean and hence how much remains in the atmosphere to affect climate. Yet, despite its crucial role, knowledge on the respiration of plankton groups is severely limited because traditional methods cannot differentiate the respiration of constituent groups within the plankton community. The reduction of the iodonitrotetrazolium salt (INT) to formazan, which when converted to oxygen consumption (O<sub>2</sub>C) using an appropriate conversion equation, provides a proxy for both total and size fractionated plankton respiration. However, the method has not been thoroughly tested with prokaryoplankton. Here we present respiration rates, as O<sub>2</sub>C and formazan formation (INT<sub>R</sub>), for a wide range of relevant marine prokaryoplankton including the gammaproteobacteria *Halomonas venusta*, the alphaproteobacteria *Ruegeria pomeroyi* and *Candidatus Pelagibacter ubique* (SAR11), the actinobacteria *Agrococcus lahaulensis*, and the cyanobacteria *Synechococcus marinus* and *Prochlorococcus marinus*. All species imported and reduced INT, but the relationship between the rate of O<sub>2</sub>C and INT<sub>R</sub> was not constant between oligotrophs and copiotrophs. The range of measured O<sub>2</sub>C/INT<sub>R</sub> conversion equations equates to an up to 40-fold difference in derived O<sub>2</sub>C. These results suggest that when using the INT method in natural waters, a constant O<sub>2</sub>C/INT<sub>R</sub> relationship cannot be assumed, but must be determined for each plankton community studied.

#### 1 Introduction

The marine biological carbon pump moves atmospheric carbon dioxide (CO<sub>2</sub>) into the ocean and ultimately sequesters carbon into the seabed. How much carbon is taken up and how much is emitted back to the atmosphere is controlled by the balance of marine photosynthesis and respiration. Therefore, knowing how much CO<sub>2</sub> is produced from prokaryoplankton, zooplankton and phytoplankton respiration is crucial to understand and predict future climate (Henson et al., 2024). In

particular, heterotrophic prokaryotes are the main organisms that remineralise dissolved and particulate organic matter during respiration and therefore are considered the main CO<sub>2</sub> producers (Robinson, 2008). However, despite the important role of these prokaryotes there is no direct method that can measure prokaryoplankton respiration (Robinson, 2019). Measurements of respiration are possible from, for example, oxygen consumption during dark incubations. However, this method cannot separate the contribution of prokaryoplankton, represented by the size fraction <0.8 µm (Aranguren-Gassis et al., 2012), from the total community respiration without creating artifacts such as removal of grazers and leaching of dissolved organic material (DOM) during the pre-incubation filtration (Aranguren-Gassis et al., 2012; Martínez-García et al., 2013).

40

60

Since respiration requires activity of the cellular electron transport system (ETS), measuring ETS activity provides an estimation of respiration. Tetrazolium salts, which act as artificial electron acceptors in the ETS, have long been used as indicators of microbial ETS activity (Packard, 1971). The reduction of the soluble 2-para (iodophenyl)-3 (nitrophenyl)-5(phenyl) tetrazolium chloride salt (INT) forms insoluble red-purple coloured formazan (INT<sub>f</sub>) crystals which can be detected spectrophotometrically. However, the classical ETS method has the limitation that it is a proxy for the maximum or potential rate of plankton respiration. It needs to be converted into actual respiration, and the empirically determined ratio between oxygen consumption and potential ETS activity is not constant (Arístegui and Montero, 1995; Hernández-León and Gómez, 1996). To circumvent this problem, Martínez-García et al., developed the INT method to estimate in vivo ETS respiration in natural marine microplankton samples (Martínez-García et al., 2009). These authors showed a constant O<sub>2</sub> consumption to formazan production ratio (O<sub>2</sub>C/INT<sub>R</sub>) of a nonaxenic algal culture and tested this ratio on a wide range of marine microbial planktonic communities (Martínez-García et al., 2009). The method was also used to estimate the contribution of the component size classes of microorganisms to the total microbial community respiration (Teira et al., 2010). Since then, the technique has been used to measure size-fractionated community respiration in a range of coastal, oligotrophic and mesopelagic environments (Aranguren-Gassis et al., 2012; Martínez-García and Karl, 2015; Teira et al., 2015; Martínez-García, 2016; García-Martín et al., 2017; García-Martín et al., 2019c).

However, the method has some unquantified limitations which must be addressed before it can be used with confidence. The original proposition of a single O<sub>2</sub>C/INT<sub>R</sub> relationship for all plankton communities and environmental conditions (Martínez-García et al., 2009) may not be appropriate because INT reduction progressively impedes ETS activity (Villegas-Mendoza et al., 2015, 2019; Baños et al., 2020), because INT may not only be reduced in the electron transport system (Maldonado et al., 2012) and because not all plankton cells necessarily take up and reduce INT at the same rate. This is exemplified by recent studies which have derived different relationships between INT reduction and O<sub>2</sub> consumption to that of Martínez-García et al. (Martínez-García et al., 2009; García-Martín et al., 2019b; García-Martín et al., 2019c). A systematic study of formazan production in representative marine plankton is therefore required. The first of these studies focussed on eukaryotic plankton species and showed that a single INT reduction to oxygen consumption conversion equation is appropriate for eukaryotic microplankton and toxicity was not reached during incubation times. However, the O<sub>2</sub>C/INT<sub>R</sub>

relationship of these eukaryote cultures was significantly different from the latest database of O<sub>2</sub>C/INT<sub>R</sub> measurements made in natural plankton communities (García-Martín et al., 2019a; García-Martín et al., 2019b).

Here we aim to validate the underlying assumptions of the INT method within the prokaryotic community (i.e. that all prokaryotes can take up INT, INT toxicity does not occur so fast as to negate an incubation of a few minutes or hours, and formazan production can be converted into oxygen consumption using a single regression equation) in order to confidently use respiration measurements derived from  $INT_R$  to evaluate the variability of prokaryote respiration in natural waters.

Six representatives of the most relevant and diverse marine prokaryotes in oligotrophic and eutrophic environments (Koch, 2001) were examined for their ability to reduce INT. Candidatus Pelagibacter ubique is a chemoheterotrophic  $\alpha$ proteobacteria of the SAR11 clade that dominates surface marine planktonic communities, especially in oligotrophic gyres (Kubo et al., 2002; White et al., 2019). C. Pelagibacter ubique has a distinctive metabolism, is minimal in size (0.019 - 0.039) μm<sup>3</sup> and 0.1 - 0.2 μm diameter approx.) and complexity, and able to oxidise low-molecular-weight dissolved organic matter (Schwalbach et al., 2010; Giovannoni, 2017; Morris et al., 2002). The autotrophs *Prochlorococcus* and *Synechococcus* are the most abundant cyanobacterial genera and the world's dominant marine photosynthetic primary producers (Mella-Flores et al., 2012). The smaller size of *Prochlorococcus* (~0.6 µm) favours its numerical dominance in the oligotrophic ocean (Rocap et al., 2003; Gomez-Pereira et al., 2013; Flombaum et al., 2013), while Synechococcus (~1 µm) can succeed in moderately oligotrophic waters (Domínguez-Martín et al., 2022b). Actinobacteria is one of the largest phyla of marine prokarvotes which can dominate deep sea sediments (Ribeiro et al., 2023). One genus within this phyla, heterotrophic Agrococcus is globally distributed (White et al., 2018), is the model member of the Actinobacteria with a mean cell size of ~1 µm (Mayilraj et al., 2006) and is regularly isolated from aquatic samples including seawater (Liu et al., 2018). Halomonas venusta is a heterotrophic and moderately halophilic γ-proteobacterium (~1.5 μm) of the Oceanospirillales which are also common in seawater (Todd et al., 2011). Ruegeria pomeroyi is a heterotrophic α-proteobacteria (~1 μm) and the model member of the Roseobacter clade (Christie-Oleza et al., 2015; Kaur et al., 2018), Roseobacters represent 20-25 % of the marine surface ocean microbial communities and have the metabolic versatility to use a wide variety of organic substrates of high molecular weight (Kaur et al., 2018; Wagner-Dobler and Biebl, 2006).

In this study we test whether these species, representatives of the key marine prokaryoplankton groups, can all reduce INT, if the method is sensitive enough to measure their respiration after an incubation time shorter than the time by which toxicity is measurable and if the rate of  $INT_R$  is consistent between prokaryote groups such that a single  $O_2C/INT_R$  conversion relationship can be used.

#### 2 Material and methods

## 95 **2.1 Growth conditions**

90

Pure species cultures were grown aerobically at their species-specific temperature (20 - 22 °C) in a 12 h light/dark cycle and were maintained at their optimum growth conditions (Moore et al., 2007; Carini et al., 2013; Wilson et al., 1996; Baumann et

al., 1972). *Halomonas venusta* HTNK1, *Ruegeria pomeroyi* DSS-3 and *Agrococcus lahaulensis* LZB509 were grown in glass Erlenmeyer flasks using minimum basal media (MBM) containing NH<sub>4</sub>Cl:K<sub>2</sub>HPO<sub>4</sub> [6:1], FeEDTA [60 mM], succinate [5 mM] and 0.1 % vitamins under ~90 μmol Q m<sup>-2</sup> s<sup>-1</sup> white light. Growth was monitored by measuring optical density at 600 nm (OD600) every few hours. *Candidatus Pelagibacter ubique* HTCC7211, the cyanobacteria *Synechococcus marinus* WH7803 and *Prochlorococcus marinus* MED4 were grown in polycarbonate Nalgene© flasks using media based on ASM1 (Carini et al., 2013), SN-ASW (Wilson et al., 1996) and PCR-S11 (Rippka et al., 2000) respectively. ASM1 was amended with pyruvate [100μM], glycine [50μM], methionine [10μM] and SAR-11 specific vitamins (B1, B5, B7, B12 and pyrroloquinoline quinone). *C. Pelagibacter ubique* and *Synechococcus m.* received ~10 μmol photons m<sup>-2</sup> s<sup>-1</sup> blue light (Moore et al., 2007; Wilson et al., 1996). Cyanobacterial growth was monitored by OD600, while growth of *C. Pelagibacter ubique* was monitored using flow cytometry (CytoFLEX Beckman & Coulter, United States) every two days. Experiments were performed when the cultures were in the exponential growth phase at cell abundances between 10<sup>4</sup> - 10<sup>7</sup> cell mL<sup>-1</sup> determined by enumeration of colony forming units (CFU) or flow cytometry (FC). To test for axenity, the cultures were gated by polygons during flowcytometry counts and streaked on agar plates and no other cells were found.

## 2.2 Respiration experiments

Each experiment was undertaken on 4 L of culture and 4 L of the appropriate media (without cells) as a negative control. The culture and control were kept in the same conditions of temperature and darkness during the experiment. Very gentle stirring ensured the homogeneity of the 4 L of culture before sampling.

Prokaryote respiration was determined from both the decrease in dissolved oxygen (Robinson et al., 2002; Serret et al., 2015) and the production of formazan (Martínez-García et al., 2009) after minutes to hours of a dark incubation. Each respiration experiment was undertaken twice for each species.

#### 2.2.1 Oxygen consumption

Dissolved oxygen was determined as the mean concentration of quadruplicate 55 mL samples collected in glass bottles and analysed by automated Winkler titration using a Metrohm 765 Dosimat titrator to a photometric end point (Carpenter, 1965; Langdon, 2010). O<sub>2</sub> consumption derived from these Winkler titrations (O<sub>2</sub>C<sub>w</sub>;  $\mu$ mol L<sup>-1</sup> h<sup>-1</sup>) was calculated from the decrease of the mean (n = 4) concentration at four-time points (O<sub>2</sub>C<sub>w</sub> = dO<sub>2</sub>/dt) during the experiment resulting in three respiration values per experiment: O<sub>2</sub>C<sub>w1</sub> = (O<sub>2</sub>)<sub>t1-t2</sub>/t<sub>1</sub>.t<sub>2</sub>, O<sub>2</sub>C<sub>w2</sub> = (O<sub>2</sub>)<sub>t2-t3</sub>/t<sub>2</sub>.t<sub>3</sub>, and O<sub>2</sub>C<sub>w3</sub> = (O<sub>2</sub>)<sub>t3-t4</sub>/t<sub>3</sub>.t<sub>4</sub>. To monitor the continuous decrease in dissolved oxygen and ensure that the experiment did not reach hypoxic conditions, an oxygen optode (NeoFox, Ocean Insight) was submerged in each of the 4 L culture and control flasks. These optode measurements were then used to calculate the consumption of dissolved oxygen from the decrease of the mean concentration of three consecutive measurements (n = 3; 1 Hz) at four time points as in the O<sub>2</sub>C<sub>w</sub> experiments (O<sub>2</sub>C<sub>opt</sub> = dO<sub>2</sub>/dt). The oxygen optodes were

calibrated using the manufacturer's recommended 2-point calibration (zero-oxygen was achieved using N<sub>2</sub> gas-saturated artificial seawater and oxygen-saturation achieved using air-bubbled artificial seawater). The experiments always ended before the culture became oxygen limited, with a mean O<sub>2</sub> saturation at the end of the experiments of 88 %.

#### 2.2.2 Formazan production

155

160

Formazan production, INT<sub>R</sub>, was determined from the difference between the mean formazan (INT<sub>f</sub>) concentration of triplicate samples of the culture and the mean of duplicate killed controls (INT<sub>c</sub>) (Martínez-García et al., 2009). Triplicate culture samples and duplicate controls were inoculated with 2-(4-iodophenyl)-3-(4-nitrophenyl)-5-phenyl tetrazolium chloride (INT) 0.2 mM final concentration and incubated for between 5 and 20 minutes, based on the optimum incubation time prior to INT induced toxicity obtained from time series experiments used for each culture (Table 1). The incubation was 140 ended by adding formaldehyde (2 % w/v final concentration). The culture-controls (20 - 30 mL) were fixed with formaldehyde (2 % w/v final concentration) for at least 20 minutes before INT was added (Martínez-García et al., 2009). The fixed samples were then filtered through 0.1 um pore size polycarbonate filters in the case of C. Pelagibacter ubique due to their smaller size ( $\varnothing \sim 0.1 - 0.2 \,\mu m$ ) and  $0.2 \,\mu m$  for the rest of the species that ranged between approximately 0.5 and 1.2  $\mu m$ in cell diameter (Schober et al., 2025; Mella-Flores et al., 2012), and stored frozen at -80 °C until analysis within a month. 145 Formazan was extracted in 1 mL propanol and the concentration calculated from the absorbance (Abs) at a wavelength of 485 nm using a Perkin Lambda 25 spectrophotometer. The formazan concentration was obtained from the calibration line (INT<sub>f</sub> = Abs \* 47.333 - 0.0141;  $R^2 = 0.9999$ ; n = 13) of commercially available pure formazan, dissolved in propanol at a range of concentrations (0.02 to 42 µmol L<sup>-1</sup>). The spectrophotometric signal of the control was removed from the samples to account for any background colour (e.g., Synechococcus m. culture is naturally reddish) or potential abiotic reduction 150 (Maldonado et al., 2012).

The INT<sub>f</sub> concentration was measured at the same three time-intervals at which the respiration experimental data  $(O_2C_{w1}, O_2C_{w2}, O_2C_{w3})$  were calculated as described above. Formazan production (INT<sub>R</sub>; µmol L<sup>-1</sup> h<sup>-1</sup>) was derived from the difference between the mean (n = 3) formazan concentration and the mean control (n = 2) and normalised to the incubation time (t = 5 - 20 min) resulting in three respiration values per experiment: INT<sub>R1</sub> = (INT<sub>f</sub>- INT<sub>c</sub>)<sub>t1</sub>/t, INT<sub>R2</sub> = (INT<sub>f</sub>- INT<sub>c</sub>)<sub>t2</sub>/t, and INT<sub>R3</sub> = (INT<sub>f</sub>- INT<sub>c</sub>)<sub>t3</sub>/t. Samples of media (without cells) as a negative control were treated in the same way as the cultures.

The optimum incubation time for each culture was estimated based on García-Martín et al. (2019a) using a non-linear least square model (a\*(1-exp(-b\*x)) with a MATLAB script (See Appendix A). Briefly, these time series experiments consist of triplicate 30 mL samples taken at each of ten-time points (e.g., t<sub>0</sub>, t<sub>1</sub>, t<sub>2</sub>...t<sub>9</sub>) from the 4 L of culture. Triplicate controls (t<sub>0</sub>) and the remaining 27 samples (t<sub>1</sub>, t<sub>2</sub>...t<sub>9</sub>) were treated and measured as described above for the estimation of INT<sub>R</sub>. The continuous decrease in dissolved oxygen was monitored with the optodes for the duration of the time series

experiments (2 - 5 h). The consumption of dissolved oxygen ( $O_2C_{opt}$ ) was calculated as described above from the decrease of the mean (n = 3) of dissolved oxygen concentration measured at these ten-time points during the time series.

INT substitutes for oxygen as the terminal electron acceptor in the cellular electron transport system, so it is progressively 'toxic' to the organism. This toxicity can occur at different times for different species and is observed as a decrease in the rate of formazan production. The increase in INT<sub>f</sub> concentration is initially linear with time but declines after a certain time  $t_x$  due to this toxic effect (Villegas-Mendoza et al., 2015; Baños et al., 2020). For each culture, the time  $t_x$  at which the toxic effect was measurable was estimated from the deviation of the exponential model to the linear model calculated from the first three data points (Fig. 1). The confidence interval of the linear regression was defined as the slope  $\pm$  one third of the standard error of the slope (García-Martín et al., 2019a). The optimum incubation time for each species was derived from the results of the time series experiments and was always  $\leq$   $t_{(x-1)}$ , to ensure that there was no bias in the experimental results due to toxicity. Each time series was repeated twice per species (Appendix A).

Figure 1. INT toxicity curve of *Agrococcus lahaulensis* during a time series experiment. A) Average INT<sub>f</sub> ( $\mu$ mol L<sup>-1</sup>) and standard error bars (SE; n = 3) are presented in black. The red line is the predicted linear fit, the red shading indicates the confidence interval, and the black line denotes the exponential rise model. B) The small top box represents a zoom-in showing toxicity after 30 minutes of incubation (vertical dashed line) and the downwards pointing arrow indicates the calculated experimental incubation time.

Respiration experiments and time series experiments were always incubated in darkness at their species-specific temperature  $\pm$  0.25 °C for the three techniques (Winkler, optodes, and INT). The calculation of respiration from each of the three techniques was always from samples taken at the same time or immediately one after the other to ensure comparable abundance and physiological conditions. The media controls from the respiration experiments and time series experiments showed no detectable oxygen consumption or formazan formation (INT<sub>f</sub> limit of detection of 0.0002  $\mu$ mol L<sup>-1</sup> h<sup>-1</sup>)

# 2.3 Respiration rates per cell

To calculate the mean and standard deviation of cell specific respiration rates per species (mean  $\pm$  SD; n = 12), we used the three respiration values INT<sub>R</sub> or O<sub>2</sub>C (O<sub>2</sub>C<sub>w</sub> and O<sub>2</sub>C<sub>opt</sub>) from the experiments and the first three INT<sub>R</sub> or O<sub>2</sub>C data points from the time series both of which were performed twice per species. The respiration rates were divided by the cell abundance determined by FC analysis (*C. Pelagibacter ubique, Synechococcus m.* and *Prochlorococcus m.*) or CFU (*Halomonas v., Ruegeria p.*, and *Agrococcus l.*). For one *Halomonas venusta* experiment, cell abundance (4.5 x 10<sup>4</sup> cell mL<sup>-1</sup>) was only estimated from optical density (OD600) calibrated against CFU versus OD600 from another *H. venusta* experiment. The relatively high uncertainty of the OD600 method to measure cell abundance compared with the accuracy of direct CFU or FC meant that we did not use this one experiment to calculate cell specific respiration rates.

# 195 **3 Results**

#### 3.1 Species-specific incubation time

All the tested prokaryoplankton took up INT and produced  $INT_f$ , but with a different time-tolerance to the toxicity of INT (Table 1 and Appendix A). Formazan was detectable from the first 5 minutes of incubation and was linear for at least the first 4 time points. This confirms that the INT method was sensitive enough to measure respiration in these cultures before toxicity was measurable.

Agrococcus lahaulensis showed the earliest toxicity (20 - 30 min) (Table 1) of all the species tested, with almost no new formazan formation after 90 minutes (Appendix A) (Fig. 1). The rate of formazan formation in *Halomonas venusta* decreased after ~ 30 minutes, although new formazan continued to be formed for the entire length of the experiment (4 h) (Appendix A). The same was observed with *Ruegeria pomeroyi*: formazan formation continued throughout the experiment, but at a lower rate after 80 - 90 minutes (Table 1) (Appendix A). *C. Pelagibacter ubique* showed marked toxicity after 30 - 41 minutes, with almost no formazan formation after one hour, showing a similar toxicity response to *Agrococcus lahaulensis* (Appendix A).

Prochlorococcus marinus showed toxicity after 60 - 82 minutes, although formazan formation continued for the entire experiment at a lower rate. Synechococcus marinus did not show any clear toxicity effect for up to 5 hours (Appendix A).

Experimental incubation times were always before toxicity was detected in the corresponding time series experiment (e.g. Fig. 1), and these ranged from 5 to 20 minutes (Table 1).

Agrococcus lahaulensis showed the highest respiration rates derived from either INT<sub>R</sub> (4.20 μmol L<sup>-1</sup> h<sup>-1</sup>) or O<sub>2</sub>C measured with either optodes (O<sub>2</sub>C<sub>opt</sub>) or the Winkler method (O<sub>2</sub>C<sub>w</sub>) (55.27 and 48.91 μmol L<sup>-1</sup> h<sup>-1</sup> respectively) (Table 1). INT<sub>R</sub> rates for *Halomonas v.*, *Ruegeria p.*, *Synechococcus m.* and *Prochlorococcus m.* ranged from 0.05 to 0.92 μmol L<sup>-1</sup> h<sup>-1</sup>, while O<sub>2</sub>C ranged from 1.08 to 25.90 μmol L<sup>-1</sup> h<sup>-1</sup>. *C. Pelagibacter ubique* showed, on average, the lowest respiration rates (INT<sub>R</sub> 0.13 μmol L<sup>-1</sup> h<sup>-1</sup>; O<sub>2</sub>C 0.45 μmol L<sup>-1</sup> h<sup>-1</sup>).

Overall, O<sub>2</sub>C<sub>opt</sub> and O<sub>2</sub>C<sub>w</sub> agreed well (R<sup>2</sup> = 0.95; p < 0.01) (Table 1) and INT<sub>R</sub> and O<sub>2</sub>C follow the same trends within each experiment (see further explanation in section 3.3: Relationship between formazan production and oxygen consumption below). The respiration rates from the two experiments undertaken for each species were not always of the same magnitude due in part to the differences in cell abundance. Therefore, we also calculated cell specific respiration rates (see following section 3.2: Single cell prokaryoplankton respiration).

**Table 1.** INT toxicity and incubation time (min), cell abundance (cell mL<sup>-1</sup>), mean and standard error (SE) of formazan production (INT<sub>R</sub>; n = 3), oxygen consumption measured with optodes (O<sub>2</sub>C<sub>opt</sub>; n = 3) and with Winkler titrations (O<sub>2</sub>C<sub>w</sub>; n = 4) in each experiment. Each species' time series and experiment (Exp. #) were performed twice (Appendix A).

| Species                 | Exp. (#) | Toxicity (min) | Incubation (min) | Abundance (cell mL <sup>-1</sup> ) | $\begin{array}{c} INT_R \\ (\mu mol~L^{\text{-}1}~h^{\text{-}1}) \end{array}$ | $\begin{array}{c} O_2C_{opt} \\ (\mu mol~L^{\text{-}1}~h^{\text{-}1}) \end{array}$ | $\begin{array}{c} O_2C_w\\ (\mu mol\ L^{\text{-}1}\ h^{\text{-}1}) \end{array}$ |
|-------------------------|----------|----------------|------------------|------------------------------------|-------------------------------------------------------------------------------|------------------------------------------------------------------------------------|---------------------------------------------------------------------------------|
| Halomonas venusta       | 1        | 29             | 11               | 1.0E+06                            | $0.62\pm0.08$                                                                 | 24.03±0.26                                                                         | 19.15±3.21                                                                      |
| HTKN1                   | 2        | 30             | 9                | 4.5E+04                            | $0.05 \pm 0.03$                                                               | 7.79±1.23                                                                          | 12.07±6.96                                                                      |
| Ruegeria pomeroyi       | 1        | 80             | 6                | 2.3E+06                            | $0.92 \pm 0.09$                                                               | 25.90±4.23                                                                         | 13.88±6.03                                                                      |
| DSS-3                   | 2        | 90             | 6                | 3.2E+05                            | $0.17 \pm 0.01$                                                               | $4.98 \pm 0.18$                                                                    | 5.87±3.23                                                                       |
| Agrococcus lahaulensis  | 1        | 20             | 6                | 2.6E+07                            | 4.20±0.11                                                                     | 55.27±0.35                                                                         | 48.91±1.17                                                                      |
| LZB509                  | 2        | 30             | 5                | 2.7E+06                            | $1.05 \pm 0.27$                                                               | 7.66±0.19                                                                          | 5.01±1.10                                                                       |
| C. Pelagibacter ubique  | 1        | 30             | 20               | 1.1E+07                            | $0.13\pm0.00$                                                                 | 0.55±0.07                                                                          | 0.61±0.19                                                                       |
| HTCC7211                | 2        | 41             | 20               | 1.5E+07                            | $0.14 \pm 0.01$                                                               | $0.31 \pm 0.05$                                                                    | 0.31±0.02                                                                       |
| Synechococcus marinus   | 1        | $\infty$       | 20               | 1.8E+07                            | $0.75\pm0.11$                                                                 | $3.73 \pm 0.44$                                                                    | 3.89±0.39                                                                       |
| WH7803                  | 2        | $\infty$       | 20               | 2.5E+07                            | $0.17 \pm 0.02$                                                               | $1.76 \pm 0.07$                                                                    | 2.19±0.64                                                                       |
| Prochlorococcus marinus | 1        | 82             | 20               | 2.0E+07                            | 0.22±0.03                                                                     | 1.60±0.30                                                                          | 1.73±0.14                                                                       |
| MED4                    | 2        | 60             | 20               | 1.3E+07                            | $0.21 \pm 0.02$                                                               | $1.08 \pm 0.10$                                                                    | 1.87±0.55                                                                       |

## 3.2 Single cell prokaryoplankton respiration

Single cell prokaryoplankton respiration was calculated from INT<sub>R</sub> and O<sub>2</sub>C (Fig. 2), for each species cultured under laboratory conditions. The experiments were performed with cell abundances between 10<sup>4</sup>-10<sup>7</sup> cell mL<sup>-1</sup>.

A symmetrical distribution of single cell respiration (median and mean are close) measured as the rate of  $INT_R$  was observed in all the experiments, apart from those with *Synechococcus m*. in which the mean cell respiration rate was 0.06

fmol h<sup>-1</sup> cell<sup>-1</sup> and the median was 0.03 fmol h<sup>-1</sup> cell<sup>-1</sup> (Fig. 2). *Halomonas v.* and *Ruegeria p.* showed the highest cell specific respiration rates (mean  $\pm$  SD: 0.57  $\pm$  0.09 and 0.57  $\pm$  0.16 fmol h<sup>-1</sup> cell<sup>-1</sup> respectively) followed by *Agrococcus l.* (0.23  $\pm$  0.14 fmol h<sup>-1</sup> cell<sup>-1</sup>). The higher s.d. of *Ruegeria p.* and *Agrococcus l.* is due to clump forming characteristics of these cultures. *Prochlorococcus m., Synechococcus m.* and *C. Pelagibacter ubique* had respiration rates one order of magnitude lower than this, with *Prochlorococcus m.* and *C. Pelagibacter ubique* having the lowest cell specific respiration rates (0.02  $\pm$  0.01 fmol h<sup>-1</sup> cell<sup>-1</sup>) (Fig. 2). Overall, the copiotrophs (*Halomonas venusta, Ruegeria pomeroyi* and *Agrococcus lahaulensis*) showed the highest rates of INT<sub>R</sub> and the oligotrophic *C. Pelagibacter ubique* and cyanobacteria (*Synechococcus marinus* and *Prochlorococcus marinus*) the lowest. Single cell rates of O<sub>2</sub>C followed the same pattern as single cell rates of INT<sub>R</sub> with *Halomonas v., Ruegeria p. and Agrococcus l.* having the highest respiration rates and *Synechococcus m., Prochlorococcus m.* and *C. Pelagibacter ubique* having the lowest rates (Fig. 2). Therefore, we separated our data into two groups based on either high or low respiration rates termed "copiotrophs" and "oligotrophs" respectively.

Figure 2. Cell specific rates of formazan production (INT<sub>R</sub> (fmol h<sup>-1</sup> cell<sup>-1</sup>) and oxygen consumption (O<sub>2</sub>C<sub>opt</sub> and O<sub>2</sub>C<sub>w</sub> (fmol h<sup>-1</sup> cell<sup>-1</sup>)); n = 12. of *Prochlorococcus marinus*, *Synechococcus marinus*, *C. Pelagibacter ubique*, *Agrococcus lahaulensis*, *Ruegeria pomeroyi* and *Halomonas venusta*. The boxplot central mark (—) indicates the median, the circle the mean (•), the side edges of the box indicate the 25th and 75th percentiles, respectively, the whiskers (¬) extend to the most extreme data points, and the outliers are plotted using the (•) symbol. Copiotrophs are shown with warm colours (dark red, orange, yellow) and oligotrophs with cold colours oligotrophs (cyan, green, violet).

# 3.3 Relationship between formazan production and oxygen consumption

The relationship between formazan production and oxygen consumption was calculated for each species from INT<sub>R</sub> versus O<sub>2</sub>C<sub>opt</sub> during time series and experiments (n = 12) and INT<sub>R</sub> versus O<sub>2</sub>C<sub>w</sub> during the experiments (n = 6). Since there were no significant differences between O<sub>2</sub>C<sub>opt</sub> and O<sub>2</sub>C<sub>w</sub> (R<sup>2</sup> = 0.95; p < 0.01), we used both datasets to calculate the relationship between formazan production and oxygen consumption. The copiotrophs *Halomonas venusta*, *Ruegeria pomeroyi* and *Agrococcus lahaulensis* all showed a good linear fit between INT<sub>R</sub> and O<sub>2</sub>C (R<sup>2</sup> = 0.75, 0.83 and 0.86, respectively) (Fig. 3). Whilst *C. Pelagibacter ubique*, *Synechococcus marinus* and *Prochlorococcus marinus* showed much lower R<sup>2</sup> values (0.33, 0.41. and 0.21, respectively). However, they were all significantly correlated (p ≤ 0.05). The lower goodness of fit was probably due to the lower range in respiration values for the oligotrophic species. The linear regression models of each species were: *Halomonas v.*: y = 0.46x + 1.52, *Ruegeria p.*: y = 0.85x + 1.35, *Agrococcus l.*: y = 1.21x + 0.87, *C. Pelagibacter ubique*: y = 0.39x + 0.04, *Synechococcus m.*; y = 0.50x + 0.55, *Prochlorococcus m.*: y = 0.29x + 0.36).

Figure 3. Logarithmic  $O_2$  consumption to formazan production ( $\mu$ mol  $L^{-1}$   $h^{-1}$ ; n = 18) linear relationship for each species. Circles represent  $O_2C$  to  $INT_R$  ( $\mu$ mol  $L^{-1}$   $h^{-1}$ ) for the prokaryoplankton *Halomonas venusta* (dark red), *Ruegeria pomeroyi* (orange), *Agrococcus lahaulensis* (yellow), *C. Pelagibacter ubique* (cyan), *Synechococcus marinus* (green) and *Prochlorococcus marinus* (violet). The  $O_2C/INT_R$  modelled linear relationship for each species is represented with a black line.

Since INT<sub>R</sub> and O<sub>2</sub>C were always significantly correlated (p  $\leq$  0.05), we investigated if the rate of formazan production relative to the rate of oxygen consumption was consistent between species such that a single O<sub>2</sub>C/INT<sub>R</sub> relationship could be used. An Analysis of Covariance with interaction term, indicated that the linear regression models of *Halomonas venusta, Ruegeria pomeroyi* and *Agrococcus lahaulensis* were statistically different from each other, while the regression models of *C. Pelagibacter ubique*, *Synechococcus m.* and *Prochlorococcus m.* were not (F = 8.38, d.f. = 5, p=  $1.31^{-6}$ ;  $\alpha = 0.05$ ).

Although the slope of each of the log O<sub>2</sub>C/INT<sub>R</sub> linear models was different, the consistently high intercept (1.52, 1.35 and 0.87 O<sub>2</sub>C (μmol L<sup>-1</sup> h<sup>-1</sup>); Fig. 3) showed that the data from all the copiotrophic prokaryotes (*Halomonas v.*, *Ruegeria p.*, *Agrococcus l.*) lie above the latest published linear model derived from O<sub>2</sub>C/INT<sub>R</sub> data collected in natural eutrophic and oligotrophic waters, which has an intercept of 0.44 O<sub>2</sub>C (μmol L<sup>-1</sup> h<sup>-1</sup>) (Fig. 4). The log O<sub>2</sub>C/INT<sub>R</sub> linear models of the oligotrophic and cyanobacterial prokaryotes have lower intercept values (0.04, 0.55 and 0.36 O<sub>2</sub>C (μmol L<sup>-1</sup> h<sup>-1</sup>); Fig. 3) more similar to that of the data collected in natural waters. The linear models of the two cyanobacteria: *Synechococcus m.* and *Prochlorococcus m.* and the *C. Pelagibacter ubique* linear model lie within the model for natural plankton communities (Fig. 4). Therefore, we could separate our data into two groups based on their position with respect to the natural database, with all the "copiotrophs" above and the "oligotrophs" within the O<sub>2</sub>C/INT<sub>R</sub> data of natural plankton communities.

Figure 4. Logarithmic O<sub>2</sub> consumption to formazan production (μmol L<sup>-1</sup> h<sup>-1</sup>) for each prokaryotic species (circles) and for natural plankton communities. *Halomonas venusta* (dark red), *Ruegeria pomeroyi* (orange), *Agrococcus lahaulensis* (yellow), *C. Pelagibacter ubique* (cyan), *Synechococcus marinus* (green) and *Prochlorococcus marinus* (violet). The

modelled linear relationship for the natural plankton community data (stars) (y = 0.72x + 0.44) is shown as a black line (García-Martín et al., 2019b).

## 290 4 Discussion

## 4.1 The uptake of INT by prokaryotes

The data presented here confirm that the INT method is always sensitive enough to measure respiration in an incubation time shorter than the time at which toxicity is measurable and additionally provide no evidence to suggest that marine prokaryotic species, represented by the cultures used here, cannot take up INT.

The toxicity of INT to prokaryotes has been tested previously in *Escherichia coli* and *Vibrio harveyi* (Smith and Mcfeters, 1997; Villegas-Mendoza et al., 2015, 2019), in a bacterial batch culture (Martínez-García et al., 2009), and recently in oligotrophic seawater samples (Baños et al., 2020). However, this is the first time that the INT method has been validated with a range of axenic marine prokaryotic species. Marine prokaryotes were represented by actinobacteria,  $\gamma$ -proteobacteria, two types of  $\alpha$ -proteobacteria, and two of the most abundant cyanobacterial genera.

The INT method has been used to predict plankton respiration from a linear regression model of O<sub>2</sub> consumption versus formazan production from natural seawater samples containing both eukaryotes and prokaryotes (Martínez-García et al., 2009; García-Martín et al., 2019b). One potential bias of the method is that cell wall structure could act as a physical barrier to the diffusion of INT before it reaches the electron transport system (ETS), which in eukaryotes occurs in the mitochondria (Berridge et al., 2005). García-Martín et al. (2019a) assessed whether five eukaryotic plankton species with different cell wall structures could take up INT and if so, whether the rate of formazan production relative to oxygen consumption was similar for all these different species. These authors found that the structure of the eukaryotic cell wall did not affect INT reduction, however the O<sub>2</sub>C/INT<sub>R</sub> relationship was significantly different from that obtained from natural plankton communities (García-Martín et al., 2019a; García-Martín et al., 2019b). To our knowledge, the effect of the cell wall in prokaryotes, both Gram-positive and Gram-negative, has not been tested. One study with inverted membrane vesicles and whole cells of the Gram-negative E. coli suggested that INT reduction may be limited by penetration through the cell envelope (Smith and Mcfeters, 1997). However, E. coli may not be representative of all marine prokaryotes because the rate of formazan production could vary depending on the organism (Smith and Mcfeters, 1997). Our results showed for the first time that six marine prokaryotic species all took up INT. This is despite the fact that Candidatus Pelagibacter ubique, which is part of the SAR11 clade, is known to have a distinctive metabolism specialized for the oxidation of low molecular weight compounds, such as amino acids, osmolytes, organic acids and volatile organic compounds, and uses few carbohydrates (Schwalbach et al., 2010; Giovannoni, 2017).

However, although all tested prokaryote species took up INT, not all of them reduced INT at the same rate relative to oxygen consumption such that a single O<sub>2</sub>C/INT<sub>R</sub> relationship could be used to convert INT<sub>R</sub> to O<sub>2</sub>C. This rate difference can be inferred from differences in the linear model between INT<sub>R</sub> and O<sub>2</sub>C for each species. In prokaryotes, unlike

eukaryotes, the reduction of INT is facilitated because the respiratory ETS is located in the cell membrane (Mitchell, 1961; Smith and Mcfeters, 1997). We explored the possibility that the different rates of formazan production were due to structural cell membrane differences between Gram-positive and Gram-negative prokaryotes. Gram-negative prokaryotes have three layers: the outer membrane containing lipopolysaccharide, the peptidoglycan cell wall, and the cytoplasmic or inner membrane (Fig. 5A). The outer layer of Gram-negative cells has little permeability for hydrophilic solutes, but diffusion is
facilitated by the outer membrane proteins or porins (Nikaido, 2003). The outer membrane also provides resistance to turgor pressure and other mechanical loads (Sun et al., 2022). Therefore, Gram-negative prokaryotes could accumulate more formazan crystals than Gram-positive prokaryotes before cell lysis.

Gram-positive cells are surrounded by thick layers of peptidoglycan but lack the outer membrane and therefore have a higher permeability (Silhavy et al., 2010). The absence of the outer layer in Gram-positive cells and the accumulation effect during the reduction of INT to insoluble formazan crystals that can lyse the cells (Gasol and Arístegui, 2007), could be the reason for the earlier toxicity found in the Gram-positive Agrococcus l. The Gram-negative cells all showed later toxicity than the Gram-positive cells, especially Synechococcus m, which didn't show any toxicity effect of INT (Fig 5B). The differences between the five Gram-negative prokaryotes might be expected, since the  $\gamma$ -proteobacteria,  $\alpha$ -proteobacteria and cyanobacterial porin channels diverge substantially from each other (Nikaido, 2003). Moreover, a recent study showed that many physical properties of the outer membrane remain unknown (Sun et al., 2022). Both the intercept and slope of the O<sub>2</sub>C/INT<sub>R</sub> relationship could be related to the time taken for INT to permeate the lipopolysaccharide and/or peptidoglycan.

The greater the intercept of the linear model for each species, the greater the amount of respiration or oxygen utilisation (and time) that occurs before formazan production is detectable, while the steeper the slope of the linear model, the higher the oxygen respiration rates are in relation to the rates of formazan production. Two of the Gram-negative prokaryotes *Halomonas venusta* and *Ruegeria pomeroyi* had the highest intercepts (Fig 5B). This could be related to a delay in formazan production due to the lower permeability of the lipopolysaccharide outer membrane compared to the Gram-positive membrane. However, *C. Pelagibacter ubique* is also Gram-negative but had the lowest intercept, suggesting a rapid diffusion of INT into the cell membrane. *C. Pelagibacter ubique* also showed one of the earliest toxicity times and the sharpest toxicity effect within the Gram-negative prokaryotes. This could be related to the small cell size (Giovannoni, 2017) and a mechanical toxic effect of intracellular crystalline formazan accumulation causing cell membrane damage (Berridge et al., 2005) and therefore earlier lysis in smaller than larger cells (Fig. 5A).

Figure 5. A) Main hypotheses that could explain the O<sub>2</sub>C/INT<sub>R</sub> relationship. ETS pathways (AOX, respiratory chain of cyanobacteria), cell membrane (Gram negative and Gram positive), metabolic rate, and cell size. B) Representation of the results from this study showing variability of INT toxicity, and formazan production versus oxygen consumption. Large blue arrows represent the elements that affect the O<sub>2</sub>C/INT<sub>R</sub> relationship. Double blue arrows represent the interaction between the elements that affect the O<sub>2</sub>C/INT<sub>R</sub> relationship. Species are represented with colours: *Halomonas venusta* (dark red), *Ruegeria pomeroyi* (orange), *Agrococcus lahaulensis* (yellow), *C. Pelagibacter ubique* (cyan), *Synechococcus marinus* (green) and *Prochlorococcus marinus* (violet).

Although most oxygen uptake in aerobic ocean ecosystems is attributed to energy-producing respiration, a variety of other biochemical processes consume oxygen, and have an unknown impact on the slope of the O<sub>2</sub>C/INT<sub>R</sub> linear model and thus the prediction of respiration rates from INT reduction. For example, relatively little is known about the function of the alternative oxidase (AOX), which is a variable trait among marine bacteria. It is thought to provide an alternative route for respiratory electron transport to oxygen, bypassing energy production (Dunn, 2023). AOX might serve a role in relieving redox stress by channelling excess electron flow to oxygen. Cells also have many non-respiratory oxidase enzymes that play roles in metabolism but have unknown INT reduction activity (Hayaishi, 2013; Giovannoni et al., 2021) (Fig. 5A).

The diversity of O<sub>2</sub>C/INT<sub>R</sub> relationships and the range of toxicity effects of INT may not only be an indicator of the differences in the cell wall type and/or AOX, but also of the different prokaryotic ETS pathways (Smith and Mcfeters, 1997) and/or different metabolisms (Fig. 5A). The four different media used here could favour different prokaryotic ETS pathways affecting the coupling of aerobic respiration and INT reduction and therefore be reflected in different O<sub>2</sub>C/INT<sub>R</sub> relationships. An analysis of the O<sub>2</sub>C/INT<sub>R</sub> relationship for each type of media (Appendix B) shows that the highest intercepts correspond to the MBM, middle values to SN-ASW and PCR-SII, and the lowest to ASM1. From the four types of media, only the MBM was repeatedly used with three species (Halomonas v., Ruegeria p. and Agrococcus l.), which could partially explain the differences between these copiotrophic species from the mesotrophs and oligotrophs. However, if the media were the main driver of the O<sub>2</sub>C/INT<sub>R</sub> relationship, then more similarity would be expected between the copiotrophs and more dissimilarity between the oligotrophs which were grown on three different media. This is the opposite to what was found here. Moreover, previous experiments with E. coli showed that the degree of coupling between oxygen consumption and INT reduction is not strongly affected by the use of succinate or NADH, but more due to the variable nature of prokaryotic respiratory chains (Smith and Mcfeters, 1997). For example, the respiratory chain of cyanobacteria has unique characteristics, the ETS is located in the thylakoids and the plasmalemma and unlike heterotrophic prokaryoplankton, it uses NADPH instead of NADH as electron donor (Packard, 1985). Another difference is the cytochrome b<sub>6</sub> complex (Fig. 5A). the counterpart of the cytochrome  $bc_1$  complexes of heterotrophic prokaryotes (Cramer et al., 2006). These differences could be the reason why the two cyanobacteria show a similar O<sub>2</sub>C/INT<sub>R</sub> relationship, but one that is different from that of the heterotrophic prokaryoplankton. We investigated whether the production of oxygen from autotrophic cyanobacteria could explain the later toxicity. However, we could not find evidence that synthesis of oxygen protects photosynthetic complexes such as photosystem I and II. In fact, extra oxygen is typically deleterious to cyanobacteria due to its readiness to react with electrons produced by photosystem II to form radical oxygen species. Cyanobacteria, including marine Synechococcus, have a range of mechanisms to remove oxygen including but not limited to terminal oxidases and flavodiiron proteins, and antioxidants such as glutathione and carotenoids (Lea-Smith et al., 2021).

Although *C. Pelagibacter ubique* is a heterotrophic prokaryote, the O<sub>2</sub>C/INT<sub>R</sub> relationship was closer to that of the cyanobacteria. At the same time, the cyanobacteria *Synechococcus marinus* and *Prochlorococcus marinus* which have O<sub>2</sub>C/INT<sub>R</sub> intercepts in the middle of the range of the tested species, have lower respiration rates than the copiotrophs and are adapted to moderate and low nutrient regions respectively (Moore et al., 2007; Kaur et al., 2018; Domínguez-Martín et al., 2022a).

C. Pelagibacter ubique is a member of the SAR11 clade, which are known for their small size and slow growth rates, traits thought to be adaptation to low nutrient environments. The carbon biomass of HTCC7211 cells, used in this study is 6.4 fg C cell<sup>-1</sup> (White et al., 2019); Prochlorococcus cells are estimated to range in carbon biomass from 30 - 79 fg C cell<sup>-1</sup> (Cermak et al., 2017), and Ruegeria carbon biomass is 141 fg C cell<sup>-1</sup> (Chan et al., 2012), a value near the low end of the range for fast growing, copiotrophic taxa, which can reach 500 fg C cell<sup>-1</sup> (Cermak et al., 2017). Since biomass data were not available for all the taxa we studied, we did not attempt to plot the relationship between INT<sub>R</sub>/ C biomass, but we

anticipate that the broad range of values for INT<sub>R</sub> cell<sup>-1</sup> shown in Fig. 2 is largely due to the more than 20-fold range in cell biomass, combined with differences in growth rates.

Regardless of the type of membrane, the cell specific INT<sub>R</sub> were positively related to the intercept values of the  $O_2C/INT_R$  equation ( $R^2 = 0.93$ ; p < 0.05), with the three copiotrophs having the highest intercepts, and the lowest respiration rates of SAR11 corresponding to the lowest intercept. Moreover, the  $O_2C/INT_R$  equations of the Gram-negative species are clearly separated into two groups (H. venusta and R. Pomeroyi versus C. Pelagibacter ubique, Synechococcus m. and Prochlorococcus m.) coincident with higher or lower respiration rates (Fig. 2 & 3 and Table 2). Therefore, the range of the  $O_2C/INT_R$  relationships (Fig. 5B) may be a combination of the differences in the cell wall type, AOX, ETS pathways and the range of metabolisms involved (Fig. 5A).

#### 4.2 The INT method as a proxy for plankton respiration in natural waters

The O<sub>2</sub>C/INT<sub>R</sub> relationships we measured were used to investigate whether a single O<sub>2</sub>C/INT<sub>R</sub> relationship can be used to derive prokaryoplankton respiration in natural waters. Since each species was cultivated at its optimum growth conditions, the rates of INT<sub>R</sub> were higher than those measured in natural waters with the same method (García-Martín et al., 2019b). Only the INT<sub>R</sub> rates measured for *C. Pelagibacter ubique* were within the INT<sub>R</sub> rates measured in natural conditions (García-Martín et al., 2019b). These higher rates suggest that toxicity times would also be faster in cultures than in natural waters. Previous studies found toxicity times between 30 minutes and 2 hours in bacterial batch cultures and experiments (Martínez-García et al., 2009; Baños et al., 2020), and between 30 minutes and 5 hours in bacterial assemblages in natural waters (Martínez-García and Karl, 2015; García-Martín et al., 2019b). Our results suggest that the time at which toxicity occurs can range from 20 min to 5 hours. It is very challenging to simulate natural oceanic conditions, especially oligotrophic ones, in the laboratory, therefore the results of laboratory experiments should only be extrapolated to natural populations with caution. Similar toxicity times are only expected in cultures growing at optimum conditions or in highly active natural eutrophic systems. However, while differences in growth conditions could explain differences in respiration rates and toxicity times, the use of three distinct respiration methods (INT<sub>R</sub>, O<sub>2</sub>C<sub>opt</sub> and O<sub>2</sub>C<sub>w</sub>) which are unlikely to be differentially affected by growth rates, suggests that the differences in O<sub>2</sub>C/INT<sub>R</sub> ratios between species could indicate plausible differences between these species or groups represented by these model species in nature.

García-Martín et al. (2019a) determined whether 5 eukaryotic plankton species could reduce INT. Here, we calculated the O<sub>2</sub>C/INT<sub>R</sub> relationship for each of these eukaryotic species to compare with the relationships of the prokaryotic species tested here (Table 2). While the respiration rates of four of the five eukaryotic species were within the O<sub>2</sub>C/INT<sub>R</sub> relationship of natural plankton communities (García-Martín et al., 2019b), the data for *Scripsiella* sp. lay above this line (Fig. 6). The respiration rates of three of the six prokaryotic species tested here, the copiotrophs, also lay above the linear O<sub>2</sub>C/INT<sub>R</sub> relationship derived from natural plankton communities.

Due to the significant metabolic differences between the prokaryotic copiotrophs and oligotrophs, we grouped our data into these two groups despite the linear O<sub>2</sub>C/INT<sub>R</sub> relationships for the three prokaryotic copiotrophs being statistically

different. The  $O_2C/INT_R$  relationships for the copiotrophs and oligotrophs are y = 0.36x + 1.21;  $R^2 = 0.36$ , and y = 0.78x + 0.52;  $R^2 = 0.40$  respectively (Table 2).

**Table 2.** Linear relationships between logarithmic oxygen consumption and formazan production from natural plankton populations including eukaryotic and prokaryotic species (García-Martín et al., 2019b), eukaryotic cultures (García-Martín et al., 2019a) and prokaryotic cultures from this study divided into copiotrophs and oligotrophs.

| Population                          |                                | Linear equation | $\mathbb{R}^2$ | Reference                 |
|-------------------------------------|--------------------------------|-----------------|----------------|---------------------------|
| Natural database                    |                                | 0.72x + 0.44    | 0.69           | García-Martín et al 2019b |
| Eukaryotes                          |                                | 0.61x + 0.62    | 0.87           | García-Martín et al 2019a |
| Thalassiosira pseudonana CCMP1080/5 |                                | 0.48x + 0.44    | 0.63           |                           |
|                                     | Emiliania huxleyi RCC1217      | 0.75x + 0.87    | 0.68           |                           |
|                                     | Scrippsiella sp. RCC1720       | 0.64x + 0.94    | 0.29           |                           |
|                                     | Oxyrrhis marina CCMP1133/5     | 0.38x + 0.23    | 0.91           |                           |
| 1                                   | Pleurochrysis carterae PLY-406 | 0.68x + 0.72    | 0.82           |                           |
| Prokaryotes                         |                                |                 |                |                           |
| Copiotrophs                         |                                | 0.36x + 1.21    | 0.36           | This study                |
|                                     | Halomonas venusta HTKN1        | 0.46x + 1.52    | 0.75           |                           |
|                                     | Ruegeria pomeroyi DSS-3        | 0.85x + 1.35    | 0.83           |                           |
| A                                   | Agrococcus lahaulensis LZB509  | 1.21x + 0.87    | 0.86           |                           |
| Oligotrophs                         |                                | 0.78x + 0.52    | 0.40           | This study                |
| C. 1                                | Pelagibacter ubique HTCC7211   | 0.39x + 0.04    | 0.33           |                           |
| S                                   | vnechococcus marinus WH7803    | 0.50x + 0.55    | 0.41           |                           |
| F                                   | Prochlorococcus marinus MED4   | 0.29x + 0.36    | 0.21           |                           |

The large difference between the  $O_2C/INT_R$  equations derived from cultured 'oligotrophic' and 'copiotrophic' prokaryotes is problematic. Currently all measurements of  $INT_R$  in natural waters are converted into  $O_2C$  using the relationship derived from natural plankton populations which span oligotrophic and mesotrophic environments (likely dominated by prokaryotes and eukaryotes respectively) (García-Martín et al., 2019b). For example, a measured value of  $INT_R$  in natural waters of 0.005 µmol  $L^{-1}$   $h^{-1}$  converted to  $O_2C$  using the linear equation derived from natural plankton communities (y = 0.72x + 0.44) would be 0.06  $O_2$  µmol  $L^{-1}$   $h^{-1}$ , a similar value (0.05  $O_2$  µmol  $L^{-1}$   $h^{-1}$ ) to that obtained with the cultured 'oligotrophs' equation obtained in this study. However, if the cultured 'copiotrophs' equation were to be used, the derived  $O_2C$  would be 40 times this at 2.43 µmol  $L^{-1}$   $h^{-1}$ . This is an unrealistically high respiration rate (58 µmol  $L^{-1}$   $d^{-1}$ ) for plankton in natural marine systems, even those in high nutrient regions (<15 µmol  $L^{-1}$   $d^{-1}$ ) (Robinson and Williams, 1999;

Robinson et al., 2009; Kitidis et al., 2014). While we do not recommend the use of the O<sub>2</sub>C/INT<sub>R</sub> equations obtained in this study when working in natural waters, it illustrates the implication of the diversity of O<sub>2</sub>C/INT<sub>R</sub> relationships, in this case that using the relationship derived from natural waters may underestimate O<sub>2</sub>C in more optimum conditions. This is a similar problem to that of many conversion equations in microbial ecology e.g. the conversion from ETS to O<sub>2</sub>C (Aristegui and Montero, 1995). Martínez-García et al. (2009) calculated the O<sub>2</sub>C/INT<sub>R</sub> relationship from a nonaxenic algal culture and tested if it would also hold for field experiments in eutrophic and oligotrophic environments. The algal culture O<sub>2</sub>C/INT<sub>R</sub> ratio agreed better with the oligotrophic field samples than the eutrophic ones (Martínez-García et al., 2009; Martínez-García and Karl, 2015), similar to the results found in the current study where the O<sub>2</sub>C/INT<sub>R</sub> relationship derived from natural plankton communities aligned with the relationship exhibited by oligotrophic prokaryoplankton species such as *C. Pelagibacter ubique, Synechococcus m.* and *Prochlorococcus m.*, but not with that of the copiotrophic species. Martínez-García and Karl (2015) estimated that the use of a single O<sub>2</sub>C/INT<sub>R</sub> relationship to derive O<sub>2</sub>C would introduce an uncertainty of 16 % in the case of oligotrophic communities and a greater than 16 % uncertainty for eutrophic communities (Martínez-García et al., 2009; Martínez-García and Karl, 2015).

**Figure 6.** Logarithmic O<sub>2</sub> consumption to formazan production (μmol L<sup>-1</sup> h<sup>-1</sup>) for each prokaryotic and eukaryotic species together with that for natural plankton communities. Prokaryotic spp.: *Halomonas venusta* (dark red), *Ruegeria pomeroyi* (orange), *Agrococcus lahaulensis* (yellow), *C. Pelagibacter ubique* (cyan), *Synechococcus marinus* (green) and *Prochlorococcus marinus* (violet). Eukaryotic spp.: *Thalassiosira pseudonana* (righthand pointing triangle), *Emiliania huxleyi* (square), *Scripsiella sp.* (upward pointing triangle), *Oxyrrhis marina* (lefthand pointing triangle) and *Pleurochrysis* 

carterae (diamond) (García-Martín et al., 2019a). The modelled linear relationship for the natural plankton communities (stars) (y = 0.72x + 0.44) is represented with a black line (García-Martín et al., 2019b).

Despite uncertainties, the range of O<sub>2</sub>C/INT<sub>R</sub> relationships and toxicity times found in this study emphasize the importance of using the appropriate relationship and incubation time when deriving plankton community respiration from measurements of formazan production in natural waters. This suggests that the measurement of O<sub>2</sub>C alongside INT<sub>R</sub> to constrain the O<sub>2</sub>C/INT<sub>R</sub> relationship in each plankton community studied needs to be continued, together with microbial community composition to assess whether the current natural population O<sub>2</sub>C/INT<sub>R</sub> relationship is representative and size fractionated INT<sub>R</sub> time series experiments to confirm that the incubation time is shorter than the time at which the rate of formazan production ceases to be linear with time.

## **5 Conclusion**

These are the first measurements of the timescale of the toxic effect of INT and the relationship between O<sub>2</sub> consumption and formazan production of a representative range of marine prokaryoplankton. All of the species tested here, including SAR11, were able to take up INT, however they did not produce formazan, relative to the consumption of dissolved oxygen, nor experience the toxic effect during formazan production, at the same rate. The difference between the O<sub>2</sub>C/INT<sub>R</sub> relationship of copiotrophic and oligotrophic species under laboratory conditions suggests that the current single linear regression conversion to O<sub>2</sub>C may not be appropriate to derive prokaryoplankton respiration rates in all natural environments. The O<sub>2</sub>C/INT<sub>R</sub> linear relationships of cultured species encountered in oligotrophic environments were not statistically different from each other, while the O<sub>2</sub>C/INT<sub>R</sub> linear relationships of prokaryoplankton species dominant in eutrophic environments were different from each other. Therefore, the currently used O<sub>2</sub>C/INT<sub>R</sub> relationship derived from natural plankton communities may not be appropriate for all natural communities. Thus, we recommend the continued derivation of the O<sub>2</sub>C/INT<sub>R</sub> relationship in natural waters, particularly in eutrophic conditions where copiotrophs dominate and continued undertaking of time series experiments, to ensure that incubations with INT are not longer than the time at which toxicity occurs in the specific community found in the area of study.

# Appendix A

Figure A1. INT toxicity curve during a time series experiment of *Halomonas venusta*, *Ruegeria pomeroyi*, *Agrococcus lahaulensis*, C. *Pelagibacter ubique*, *Synechococcus marinus* and *Prochlorococcus marinus*. Average INT<sub>f</sub> (μM) and

standard error bars (SE; n = 3) are presented in black. The red line is the predicted linear fit, the red shading indicates the confidence interval, and the black line denotes the exponential rise model. The vertical dashed line represents initial signs of toxicity.

# Appendix B

Figure B1. O<sub>2</sub>C/INT<sub>R</sub> intercept and slope values found using each type of media MBM, ASM1, SN-ASW, and PCR-SII.

Data availability. Respiration rates and cell abundances are available for download at BODC doi:10.5285/2be8f599-592c-5de2-e063-7086abc02acd

Authors contributions. IS: Conceptualization, data curation, formal analysis, investigation, project administration, visualization, writing-original draft preparation, writing-review and editing. KV: Investigation, writing-review and editing. JDT: Funding acquisition, resources, writing-review and editing. SJG: Resources, writing-review and editing. EEG: Conceptualization, funding acquisition, writing-review and editing. RU: Investigation, writing-review and editing. CR: Conceptualization, funding acquisition, project administration, supervision, writing-review and editing.

Competing interests. The authors declare that they have no conflict of interest.

Acknowledgements. The authors wish to thank Chih-Ping Lee and Sarah S. Wolf from Oregon State University, David Scanlan from the University of Warwick, Michelle Michelsen and Holger H. Buchholz from the University of Exeter, Jinyan Wang from the Ocean University of China, and David Lea-Smith and Nicholas Garrard from UEA for their support and help

with the bacterial cultures. We would like to thank the two reviewers for their constructive comments which improved this manuscript.

Financial support. This study was funded by the UK Natural Environment Research Council (NERC) project "REMineralisation of organic carbon by marine bActerIoplanktoN (REMAIN) - reducing the known unknown" (NE/R000956/1) and The Leverhulme Trust project "Marine bacterioplankton respiration: a critical unknown in global carbon budgets" (RPG-2017-089) awarded to CR. During the writing of this manuscript IS was funded by the UK NERC project "The abiotic and biotic factors determining microbial respiration, a key process in ocean carbon storage (MicroRESPIRE)" (NE/X008630/1) awarded to CR as part of the BioCARBON strategic research programme.

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
