# Peer review of "Respiration rates of marine prokaryotes and implications for the in vivo INT method"

_EGUsphere, 2025_

## Referee Comment (RC2)

**1)**

**Summary**

The study demonstrates that all tested prokaryoplankton species import and reduce INT. Still, the $O_2$ consumed and INT reduced ratio is species-dependent, making it impossible to assume a single universal conversion factor.

The authors quantify species-specific toxicity and adjust incubation times (5–20 min) to avoid bias, also showing good agreement between $O_2$ measured by optodes and by Winkler titration.

In copiotrophs (*Halomonas, Ruegeria, Agrococcus*) the INTR–O2C relationship has a strong linear fit; in oligotrophs and cyanobacteria (SAR11, *Synechococcus, Prochlorococcus*), the fit is significantly weaker, but the linear model lies within the model for natural plankton communities (Fig. 4).

The conclusion is that in situ studies must derive the O2C/INTR relationship locally and check for toxicity in the studied community.

**2)**

**Scientific questions**

The study addresses the quantification of prokaryotic plankton respiration and the validity of the in vivo INT method.

**Novelty**

- Comparative multi-taxa dataset under controlled conditions.
- Derivation and comparison of slopes/intercepts per species,
- An operational framework to define toxicity and optimal incubation times.

**Conclusions**

Conclusions are well supported: no single O2C/INTR factor exists, and studies must derive species-specific relationships (especially in eutrophic systems dominated by copiotrophs).

**Methods and assumptions**

- Robust, $O_2$ measured by optodes and Winkler titration, avoiding hypoxia.
- INTR with INT 0.2 mM, killed and media controls, propanol extraction, and calibration curve.
- Quantitative toxicity criterion and choice of incubation.
- Per-cell rates calculated from FC/CFU counts.

**Support for results**

- Toxicity curves (Appendix A), per-cell rates (Appendix B with Winkler and optodes), and species-specific relationships are presented.
- Table 1 with toxicity, abundance, and O2C/INTR values from replicates per species.

**Traceability and reproducibility**

Solutions, equipment, calibrations, and regression models.
The authors declare the availability of the data in the public BODC repository and provide a DOI. However, at the time of this review, the dataset could not be accessed through the provided link. Authors are requested to verify the DOI and ensure the data are fully accessible to the public before final publication.

**Credit and originality**

Context and limitations of the method (classic ETS, constant O2C/INTR assumption) are well referenced and contrasted; the authors' contribution (species test and comparative analysis) is clearly stated.

**Title**

Clear. Respiration rates of marine prokaryotes and implications for the in vivo INT method.

**Abstract**

Complete. States the problem, approach, organisms, main finding (O2C/INTR variability), and methodological implications.

**Structure and clarity**

Introduction–Methods–Results–Discussion–Conclusions–Appendices. Figures and tables are well integrated with clear cross-referencing.

**Language**

Technical English is fluent and precise.

**Mathematical formulation and symbols**

Units are consistent, equations and parameters are defined.

**References**

Appropriate in number and quality, covering the state of the art.

**Supplementary material**

Pertinent:
Appendix A (toxicity curves) and Appendix B (per-cell $O_2$ rates).

**3) Major comments**

*Ecological generalization and growth media*

Although the discussion notes possible effects of growth medium and respiratory chain diversity, extrapolation to natural communities could be strengthened with an analysis showing how much slopes/intercepts vary across media.

*Underlying physiological mechanisms*

The discussion proposes several hypotheses (cell wall, AOX, etc.) to explain the observed variability. To unify these ideas, the creation of a conceptual diagram is recommended. This figure would serve as a visual summary, linking the proposed mechanisms with their theoretical effects on the $O_2C$ and INTR.

**4) Editorial recommendation**

- Accept with minor revisions. The manuscript provides new and relevant evidence for the *in vivo* INT method, and the conclusions are well supported and do not require new experiments.

---

## Author Comment (AC2)

**RC2**

**1) Summary**

The study demonstrates that all tested prokaryoplankton species import and reduce INT. Still, the $O_2$ consumed and INT reduced ratio is species-dependent, making it impossible to assume a single universal conversion factor.

The authors quantify species-specific toxicity and adjust incubation times (5–20 min) to avoid bias, also showing good agreement between $O_2$ measured by optodes and by Winkler titration.

In copiotrophs (*Halomonas, Ruegeria, Agrococcus*) the INTR–O2C relationship has a strong linear fit; in oligotrophs and cyanobacteria (SAR11, *Synechococcus, Prochlorococcus*), the fit is significantly weaker, but the linear model lies within the model for natural plankton communities (Fig. 4).

The conclusion is that in situ studies must derive the O2C/INTR relationship locally and check for toxicity in the studied community.

**2) Scientific questions**

The study addresses the quantification of prokaryotic plankton respiration and the validity of the in vivo INT method.

**Novelty**

- Comparative multi-taxa dataset under controlled conditions.
- Derivation and comparison of slopes/intercepts per species,
- An operational framework to define toxicity and optimal incubation times.

**Conclusions**

Conclusions are well supported: no single O2C/INTR factor exists, and studies must derive species-specific relationships (especially in eutrophic systems dominated by copiotrophs).

**Methods and assumptions**

- Robust, $O_2$ measured by optodes and Winkler titration, avoiding hypoxia.
- INTR with INT 0.2 mM, killed and media controls, propanol extraction, and calibration curve.
- Quantitative toxicity criterion and choice of incubation.
- Per-cell rates calculated from FC/CFU counts.

**Support for results**

- Toxicity curves (Appendix A), per-cell rates (Appendix B with Winkler and optodes), and species-specific relationships are presented.
- Table 1 with toxicity, abundance, and O2C/INTR values from replicates per species.

**Traceability and reproducibility**

Solutions, equipment, calibrations, and regression models.

The authors declare the availability of the data in the public BODC repository and provide a DOI. However, at the time of this review, the dataset could not be accessed through the provided link. Authors are requested to verify the DOI and ensure the data are fully accessible to the public before final publication.

**We have double checked that the link works. We also provide another link that links to the webpage were the DOI is also stated:**
**https://www.bodc.ac.uk/data/published_data_library/catalogue/10.5285/2be8f599-592c-5de2-e063-7086abc02acd/**

**Credit and originality**

Context and limitations of the method (classic ETS, constant O2C/INTR assumption) are well referenced and contrasted; the authors' contribution (species test and comparative analysis) is clearly stated.

**Title**

Clear. Respiration rates of marine prokaryotes and implications for the in vivo INT method.

**Abstract**

Complete. States the problem, approach, organisms, main finding (O2C/INTR variability), and methodological implications.

**Structure and clarity**

Introduction–Methods–Results–Discussion–Conclusions–Appendices. Figures and tables are well integrated with clear cross-referencing.

**Language**

Technical English is fluent and precise.

**Mathematical formulation and symbols**

Units are consistent, equations and parameters are defined.

**References**

Appropriate in number and quality, covering the state of the art.

**Supplementary material**

Pertinent:

Appendix A (toxicity curves) and Appendix B (per-cell $O_2$ rates).

**3) Major comments**

*Ecological generalization and growth media*

Although the discussion notes possible effects of growth medium and respiratory chain diversity, extrapolation to natural communities could be strengthened with an analysis showing how much slopes/intercepts vary across media.

**We have modified the text in the discussion Section 4.1 (line 355) "An analysis of the $O_2C/INT_R$ relationship for each type of media (Appendix C) shows that the highest intercepts correspond to the MBM, middle values to SN-ASW and PCR-SII, and the lowest to ASM1, which is coincident with eutrophic, mesotrophic and oligotrophic environments where the copiotrophs, cyanobacteria and SAR11 respectively are found. From the four types of media, only the MBM was repeatedly used with three species (*Halomonas v.*, *Ruegeria p.* and *Agrococcus l.*), which could partially explain the differences between these copiotrophic species from the mesotrophs and oligotrophs. However, if the media were the main driver of the $O_2C/INT_R$ relationship, then more similarity would be expected**

**between the copiotrophs and more dissimilarity between the oligotrophs which were grown on three different media. This is the opposite to what was found here.”**

**Added a plot showing how the intercepts and slopes vary across media (Appendix C)**

[Figure]

*Underlying physiological mechanisms*

The discussion proposes several hypotheses (cell wall, AOX, etc.) to explain the observed variability. To unify these ideas, the creation of a conceptual diagram is recommended. This figure would serve as a visual summary, linking the proposed mechanisms with their theoretical effects on the $O_2C$ and INTR.

[Figure]

**Figure X.** A) Main hypothesis of that can explain the $O_2C/INT_R$ relationship. ETS pathways (AOX, respiratory chain of cyanobacteria), cell membrane (Gram negative and Gram positive), metabolic rate, and cell size. B) Representation of the results from this study showing

variability of INT toxicity, and formazan production versus oxygen consumption. Big blue arrows represent the elements that affect $O_2C/INT_R$ relationship. Double arrows represent the interaction between the elements that affect the $O_2C/INT_R$ relationship. Species are represented with colours: *Halomonas venusta* (dark red), *Ruegeria pomeroyi* (orange), *Agrococcus lahaulensis* (clear yellow), *C. Pelagibacter ubique* (cyan), *Synechococcus marinus* (green) and *Prochlorococcus marinus* (violet).

**4) Editorial recommendation**

- Accept with minor revisions. The manuscript provides new and relevant evidence for the *in vivo* INT method, and the conclusions are well-supported and do not require new experiments.

---

## Author Comment (AC3)

RC1:

This study examined the ETS (electron transport system) method, which has gained traction as a tool for estimating respiration in marine plankton communities, with a focus on prokaryoplankton. The authors measured the $INT_R$ and oxygen consumption (Winkler titrations and optodes) simultaneously on a wide range of relevant marine prokaryoplankton to establish the empirical equations between $O_2C$ and $INT_R$. They examined whether it is constant within species and whether it can be extrapolated to natural plankton communities. Overall, this study is of significant necessity, serving as an essential reference for refining the ETS method. Also, the manuscript is well-written. I have several comments for further improving this manuscript.

**Thank you for the constructive comments and suggestions. Find here below our changes based on your comments in bold font.**

Line 15: Spell out the full name for "prokaryo-, zoo- and phytoplankton".

**Agreed**

Line 31: Also, use "prokaryoplankton, zooplankton" instead of "bacterio-, zoo-".

**Agreed**

Line 37: add reference for 0.8 μm. Some studies used 1 μm for prokaryoplankton.

**The reference is at the end of the sentence but added it again after the 0.8 μm for clarity.**

Line 137: How are the incubation times "between 5 and 20 minutes" determined?

**Explained now: "based on the optimum incubation time prior to INT induced toxicity obtained from time series experiments used for each culture"**

Line 191: use "FC" instead of flowcytometry to be consistent with previous text.

**Agreed**

Table 1: It is better to add one column to show the experiment number to distinguish the two experiments.

**Added**

Fig. 2: Add mean values to the boxplot to illustrate the data distribution more visually. Also, I suggest putting all the figures of single-cell respiration, including O2C, into Fig. 2. All the figures support the data grouping into "copiotrophs" and "oligotrophs".

**Mean values added to the boxplots. We have combined all the previous figures of single cell respiration into Figure 2 and therefore removed them from the second appendix. Grouping is already mentioned in the text and represented in the plots as warm colours**

for copiotrophs (red, orange, yellow) versus cold colours for oligotrophs (blue, green, purple), and I added this text to the figure's capture as well.

[Figure]

**Figure 2. Cell specific rates of formazan production ($INT_R$ (fmol h$^{-1}$ cell$^{-1}$) and oxygen consumption ($O_2C_{opt}$ and $O_2C_w$ (fmol h$^{-1}$ cell$^{-1}$)) ; n = 12. of *Prochlorococcus marinus*, *Synechococcus marinus*, *C. Pelagibacter ubique*, *Agrococcus lahaulensis*, *Ruegeria pomeroyi* and *Halomonas venusta*. The boxplot central mark (—) indicates the median, the circle the mean (○), the side edges of the box indicate the 25th and 75th percentiles, respectively, the whiskers (⊤) extend to the most extreme data points, and the outliers are plotted using the (•) symbol. Copiotrophs are shown with warm colours (red, orange, yellow) and oligotrophs with cold colours oligotrophs (blue, green, purple).**

Line 249: Pooling the data from both methods (i.e., Winkler titrations and optodes) to build a linear regression requires the assumption that there are no significant differences between the two methods. Otherwise, it is preferable to use a single dataset.

**We tested for significant differences and overall, we found that $O_2C_{opt}$ and $O_2C_w$ agreed very well ($R^2$ = 0.95; p < 0.01) (Table 1) and therefore decided to combine these data. This is stated two sections above (line 218), and therefore, to make it clearer, we added the following text (line 252): "Since there was no significant difference between $O_2C_{opt}$ and $O_2C_w$ ($R^2$ = 0.95; p < 0.01), we combined the datasets to construct the relationship between formazan and oxygen consumption.**

Fig. 3: Add p-value to each sub-figure.

**Added to each sub-figure**

Line 266: Which statistical method was used for the covariance analysis? Please specify it.

**The Analysis of Covariance (ANCOVA) was performed with the Matlab function aoctool, this analysis includes an interaction term, and accounts for the differences in**

**the slopes. We specified the inclusion of an interaction term in the text to clarify that is not a one-way ANCOVA (line 269)**

Fig. 5 and Fig. 4 are similar. It seems not necessary to use two figures. I suggest removing Fig. 4 and moving Fig. 5 forward.

**Although figures 4 and 5 look similar, they represent different things and removing figure 4 would make the results section unclear. Figure 4 shows the results of this study while figure 5 is part of the discussion and compares our results with previous experiments with autotrophs. We therefore would like to maintain Fig 4 to not include previous studies within the results section and include Fig 5 for clarity and better flow of the discussion.**

Line 424: Which equation was used for this calculation? Please specify it.

**($y = 0.72x + 0.44$) added to the text (line 432)**

Conclusion: Can you draw more specific findings from your study or suggestions for the experiments in natural waters? For instance, within what time frame can safety be guaranteed without triggering toxicity?

**We decided to do not suggest specific time frames in the conclusion because although our laboratory experiments show the large differences between species, they don't necessarily indicate the exact time frame for incubations in natural waters. That is why we recommend performing a time series experiment in the area of study first to find out the safe time frame for respiration incubation measurements.**

**However, we have added further explanation within the discussion, in section 4.2, line 401, we say "These higher rates suggest that toxicity times would also be faster in cultures than in natural waters. Previous studies found toxicity times between 30 minutes and 2 hours in bacterial batch cultures and experiments (Martínez-García et al., 2009; Baños et al., 2020), and between 30 minutes and 5 hours in bacterial assemblages in natural waters (Martínez-García and Karl, 2015; García-Martín et al., 2019b). Our results suggests that the time at which toxicity occurs can vary from 20 min to 5 hours. It is very challenging to simulate natural oceanic conditions, especially oligotrophic ones, in the laboratory, therefore the results of laboratory experiments should only be extrapolated to natural populations with caution. Similar toxicity times are only expected in cultures growing at optimum conditions or in highly active natural eutrophic systems."**